**Data Availability Statement:** All data are submitted as supporting information files.

**Funding:** This work is funded by the Research Evidence and Development Initiative (READ-It),

# Is routine Vitamin A supplementation still justified for children in Nepal? Trial synthesis findings applied to Nepal national mortality estimates

Samjhana Shrestha[1,2]☯, Saki Thapa[1], Paul Garner[2], Maxine Caws[1,3], Suman Chandra Gurung[1,3], Tilly Fox[2], Richard Kirubakaran[4], Khem Narayan Pokhrel[1]☯*

1 Birat Nepal Medical Trust (BNMT), READ-It Project, Kathmandu, Nepal, 2 Centre for Evidence Synthesis in Global Health, Liverpool School of Tropical Medicine, Liverpool, United Kingdom, 3 Department of Clinical Sciences, Center for Drugs and Diagnostics, Liverpool School of Tropical Medicine, Liverpool, United Kingdom, 4 Prof BV Moses Centre for Evidence-Informed Healthcare and Health Policy, Vellore, India

☯ These authors contributed equally to this work.
* pokhrelkhemn@gmail.com

## Abstract

### Background

The World Health Organization has recommended Vitamin A supplementation for children in low- and middle-income countries for many years to reduce child mortality. Nepal still practices routine Vitamin A supplementation. We examined the potential current impact of these programs using national data in Nepal combined with an update of the mortality effect estimate from a meta-analysis of randomized controlled trials.

### Methods

We used the 2017 Cochrane review as a template for an updated meta-analysis. We conducted fresh searches, re-applied the inclusion criteria, re-extracted the data for mortality and constructed a summary of findings table using GRADE. We applied the best estimate of the effect obtained from the trials to the national statistics of the country to estimate the impact of supplementation on under-five mortality in Nepal.

### Results

The effect estimates from well-concealed trials gave a 9% reduction in mortality (Risk Ratio: 0.91, 95% CI 0.85 to 0.97, 6 trials; 1,046,829 participants; low certainty evidence). The funnel plot suggested publication bias, and a meta-analysis of trials published since 2000 gave a smaller effect estimate (Risk Ratio: 0.96, 95% CI 0.89 to 1.03, 2 trials, 1,007,587 participants), with the DEVTA trial contributing 55.1 per cent to this estimate. Applying the estimate from well-concealed trials to Nepal's under-five mortality rate, there may be a reduction in mortality, and this is small from 28 to 25 per 1000 live births; 3 fewer deaths (95% CI 1 to 4 fewer) for every 1000 children supplemented.

which is an accountable grant of UK aid through the Foreign, Commonwealth and Development Office (FCDO) (Project number 300342-104). The funders had no role in study design, data collection and analysis, decision to publish, or preparation of the manuscript.

**Competing interests:** The authors have declared that no competing interests exist.

## Conclusions

Vitamin A supplementation may only result in a quantitatively unimportant reduction in child mortality. Stopping blanket supplementation seems reasonable given these data.

## Background

For over 20 years, the World Health Organization (WHO) have recommended Vitamin A supplements to all children under five years (6–59 months) in low- and middle-income countries (LMICs) [1]. Although recommendations have changed with Vitamin A supplementation (VAS) now only being recommended when more than 1% of children have night blindness, or when one-fifth of the children have low retinol levels, in practice, most governments continue with giving supplements routinely [2]. Nepal, for instance, has been distributing large amounts of supplements to children since its start in 1993 [3] despite substantial progress in the health of under-five children [4].

The global evidence that the country refers to for child survival benefits of VAS has come from the Cochrane review by Imdad et al. who conclude, "Vitamin A supplementation is associated with a clinically meaningful reduction in morbidity and mortality in children." However, the trials in the review are a decade-old trials [5,6], and were conducted against a backdrop of high childhood infections rates, greater malnutrition, and higher child mortality. On top of this, the Cochrane review has considerable unexplained heterogeneity ($I^2$ = 61%) [6] which may indeed mean that the intervention works in some circumstances, but not in others. We know that the health status in Nepal has been improved (See Table 1 for details). In this analysis, we assess whether the mortality reduction with VAS in randomized controlled trials (RCTs) translates into important health benefits today in Nepal.

### Aim and objectives

Our research question was whether routine vitamin A supplementation was still justified in Nepal? We sought to estimate the effects of routine vitamin A supplementation by a) generating an up-to-date reliable estimate of the effect of Vitamin A on mortality by updating the 2017 Cochrane meta-analysis of mortality, and then b) estimating the effects of supplementation on absolute mortality using contemporary health status measures of children.

## Methods

### Update of Cochrane review

We used the same methods from the 2017 Cochrane systematic review to update it [6]. We considered RCTs and cluster RCTs conducted among children for assessing the effect of VAS in reducing child mortality. We restricted the outcomes to all-cause child mortality. Full details of the methods are included in the supporting information ("S1 Appendix") which details our prespecified effect modifiers and sub-group analysis and intended analysis of absolute effects against current measures of child health status in Nepal.

### Search methods for identification of studies

We searched databases and trials registers from 2016 to March 2021 using the same search strategies as used in the Cochrane review [6] to identify any new relevant studies besides those included in the review.

**Table 1. Secular changes in child health and health care delivery in Nepal.**

| Secular changes | Earlier estimate | Contemporary estimate |
|---|---|---|
| Under Five Mortality Rate (U5MR) | **1996**<br>118 deaths per 1,000 live births[1] | **2019**<br>28 deaths per 1,000 live births[2] |
| Measles burden | **2003**<br>5419 cases[3] | **2019**<br>424 cases[3] |
| Prevalence of diarrhoea | **1996**<br>28%[1] | **2016**<br>8%[1] |
| Diarrhoea Case Fatality Rate | - | 2019<br>Less than 1 per 1000[3] under-five children |
| Measles immunization coverage | **1996**<br>57%[1] | **2016**<br>90%[4] |
| Vitamin A coverage | 32%[1] | 86%[4] |
| Vitamin A Deficiency (Serum retinol<0.7micromol/L) | **1998**<br>32.3%[5] | **2016**<br>12.5%[6] |
| Vitamin A Deficiency (Modified Relative Dose-Response (MRDR) >0.060) | - | **2016**<br>4.2%[6] |
| Night blindness (12–59 months) | - | **1998**<br>0.27%[5] |
| Stunting prevalence | **1996**<br>57%[1] | **2019**<br>31.5%[2] |
| Wasting prevalence | 15%[1] | 12%[2] |
| Under-weight prevalence | 42%[1] | 24%[2] |

[1] Nepal Family Health Survey 1996 [7].

[2] Multi-Indicator Cluster Survey 2019 [8].

[3] Annual Report, Department of Health Services, 2018/19 [3].

[4] Nepal Demographic Health Survey 2016 [9].

[5] Nepal Micronutrient Survey 1998 [10].

[6] Nepal National Micronutrient Status Survey 2016 [11].

## Data collection and analysis

Using the inclusion criteria noted in our systematic review protocol (CRD42021249941), we screened the trials included in the 2017 Cochrane review and the trials that resulted from a new search. We extracted data on the effect modifiers from each trial. For trials where such data are not reported, we considered estimates of such modifiers from different sources such as the Global Health Observatory data repository [12] and other potential data sources like demographic health surveys, multi-indicator cluster surveys, and the World Bank data repository [13].

## Assessment of risk of bias

We assessed the risk of bias using the Cochrane Risk of Bias Tool [14]. We improved on the risk of bias in the 2017 Cochrane review by assessing the cluster RCTs by recording baseline imbalance, loss of clusters, and the possibility of bias arising due to recruitment of participants into clusters, incorrect analysis and comparability with individually randomized trials.

## Unit of analysis issues

For studies that did report cluster adjustments process or intra-cluster correlation coefficient (ICC), we used the design effects in the previous review [15] to adjust for clustering in those trials which did not control for clustering.

## Data synthesis

We conducted a meta-analysis using a fixed-effect model in RevMan version 5.4 software [16]. When the heterogeneity observed in the analysis was substantial (p-value <0.10 and $I^2 >$ 50%), we also considered a random-effect model.

## Subgroup analysis

We considered current measures of child health status in Nepal for performing subgroup analyses. For instance, we performed subgroup analysis by background U5MR and used the cut-off criteria that reflected the current U5MR in Nepal (categorized as U5MR between 30 and 60 per 1000 live births).

## GRADE assessment

We assessed the quality of the evidence generated from the review using the approach, Grading of Recommendations Assessment, Development and Evaluation (GRADE) [17].

# Results

## Description of studies

**Results of the search.** We considered 19 studies from the 2017 Cochrane review [6] that were included in the meta-analysis of all-cause mortality. The new electronic searches conducted for the period 2016 to March 2021 only identified one article to be assessed for full-text (See Fig 1). Altogether, 20 full-text articles (19 from the previous review and 1 from the new search) were assessed for eligibility, of which, 4 studies [18–21] were excluded (See "S2 Appendix" for reasons for exclusion). We, thus, identified 16 studies (See "S3 Appendix") as eligible for inclusion in the review. However, of those 16 studies considered for inclusion, one study [22] did not report the methods well and details were insufficient to be sure this was a genuine RCT. We tried to obtain additional information but were unsuccessful, so excluded this from the analysis.

## Included studies

**Types of studies.** Of the included studies, five trials [23–27] were individual-RCT designs and the remaining ten trials [28–37] were cluster-RCT. Further information about individual studies is available in "S3 Appendix".

**Location/Setting.** Studies took place in 6 countries; five trials in India [26,28,32,35,36], three trials in Nepal [29,31,37], one trial in Indonesia [34], two trials in Ghana [25,33], two trials in Guinea-Bissau [23,24] and one trial each in Sudan [30] and Congo [27].

**Participants.** Trials in this review incorporated approximately around 1,200,679 participants, with a sample size varying from 462 [23] to up to around 1 million participants [35].

**Interventions.** All trials examined the effects of Vitamin A supplementation given to children. Vitamin A doses used in the trials ranged from 8333 IU to 200,000 IU. In general, younger age groups (<12 months) received lower doses and older age groups (>12 months) received the higher dose.

**Comparisons.** Five trials compared VAS against usual care [27,29,31,34,35] and the remainder of the ten trials used a placebo as a comparator.

**Multiple-arm trials.** Two trials had multiple arms, one combining the measles vaccine with Vitamin A and the other combining Vitamin A with deworming tablets [23,35].

**Outcomes.** All trials reported child death as the outcome and used home visits to collect such mortality data. Only one trial [24] reported side effects associated with Vitamin A supplementation.

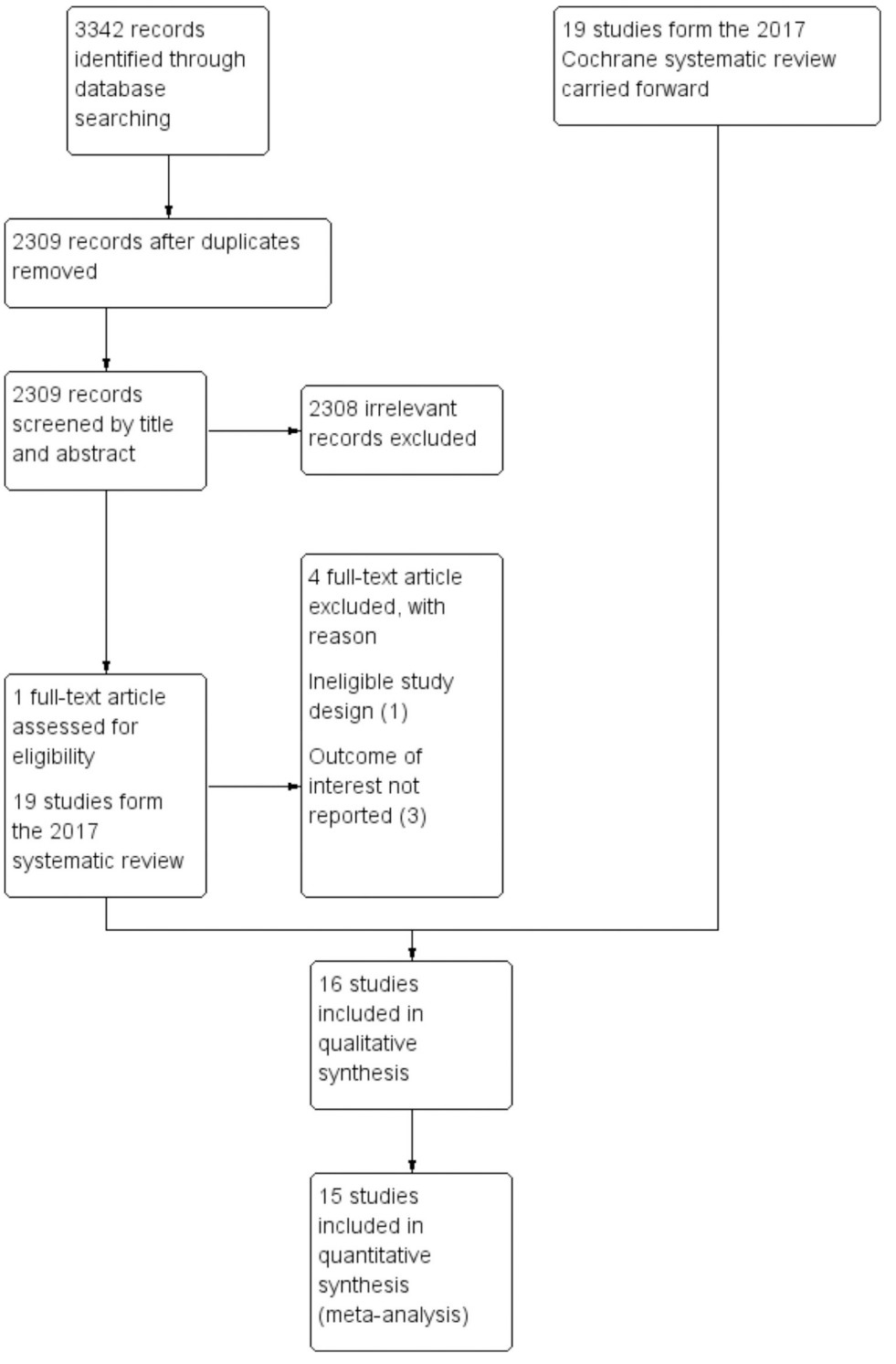

**Fig 1. Study selection flow diagram.**

**Other study characteristics.** The background parameters of the trials included in the review featured high child mortality, greater malnutrition-indicated by wasting levels, high xerophthalmia (Night blindness, Bitot's spot) among children under five years, and no vitamin

**Table 2. Comparison of the characteristics by effect modifiers in Nepal at present and characteristics of the effect modifiers in the trials.**

| Name of the study | Study Start Year | Country | U5MR | Wasting | Xerophthalmia | Measles immunization | Vitamin A coverage | Data source |
|---|---|---|---|---|---|---|---|---|
| **Current status of child health in Nepal** | | | | | | | | |
| Nepal | 2019 | Nepal | 28/1000 live births | 12% | NA | 87% | 80% | Nepal Multi-Indicator Cluster Survey 2019 [8] DoHS, 2019 [3] |
| Highly deprived province | 2019 | | 30/1000 live births | 17.6% (Karnali) | NA | 91% | 89% | Highest U5MR in Province 5: 40/1000 live births |
| Less deprived province | 2019 | | 19/1000 live births | 4.7% (Bagmati) | NA | 94% | 69% | Nepal Multi-Indicator Cluster Survey 2019 [8] |
| **Background trial characteristics by effect-modifiers** | | | | | | | | |
| Agarwal 1995 | 1990 | India | 109/1000 | 16% | 2.2% | 26.3% | 0% | National Family Health Survey 1992–93 [38] |
| Ben 1997 | 1993 | Guinea-Bissau | 203/1000 | 10% | 0.004% | 49% | 43.7% | Multi-Indicator Cluster Survey 2000 [39,40] |
| Daulaire 1993 | 1989 | Nepal | 126/1000 | 26% | 13.2% | 37% | 0% | World Bank data repository [13] |
| DEVTA trial 2013 | 1999 | India | 96/1000 | 15% | 2.55% | 38% | 6% | National Family Health Survey 2005–06 [41] |
| Donnen 1998 | 1998 | Congo | 186/1000 | 6.13% | 0.001% | 38% | 0% | World Bank data repository [13,40] |
| Fisker 2014 | 2013 | Guinea-Bissau | 98/1000 | 6% | 0 | 41.5% | 54.5% | Multi-Indicator Cluster Survey 2014 [42,43] |
| Herrera 1992 | 1988 | Sudan | 135/1000 | 6% | 2.85% | 67% | 20.05% | Demographic Health Survey 1989–90 [44] |
| Pant 1996 | 1992 | Nepal | 118/1000 | 67% | 1% | 57% | 32% | Nepal Family Health Survey 1996 [7] |
| Rahmathullah 1990 | 1989 | India | 130/1000 | 23% | 11% | 42% | 1% | World Bank data repository [13] |
| Ross 1993 Health | 1990 | Ghana | 127/1000 | 4% | 1.5% | 50% | 0% | World Bank data repository [13] |
| Ross 1993 Survival | 1989 | Ghana | 132/1000 | 7% | 0.7% | 44.5% | 0% | World Bank data repository [13] |
| Sommer 1986 | 1983 | Indonesia | 115.5/1000 | 3.5% | 2.1% | 13% | 0% | Demographic Health Survey 1991 [45] |
| Venkatrao 1996 | 1991 | India | 109/1000 | 17.5% | 2% | 72% | 0% | National Family Health Survey 1992–93 [38] |
| Vijyagharvan 1990 | 1987 | India | 130/1000 | 20% | 6% | 42% | 0% | World Bank data repository [13] |
| West 1990 | 1989 | Nepal | 126/1000 | 6% | 3% | 57% | 0% | Nepal Family Health Survey 1996 [7] World Bank data repository [13] |

A supplementation programs (See Table 2). The trials' background contrasts with the current measures of child health status in Nepal (See Table 2).

## Risk of bias in included studies

We assessed the risk of bias in the 15 studies included in the analysis and assigned them as having high, low or unclear risk. We presented the results of the risk of bias assessment across all trials in Fig 2. Sixty per cent of the included studies (9 studies) did not have sufficient information on the sequence generation and allocation concealment process and thus had an unclear risk of bias. We assessed cluster RCTs for other potential sources of bias. Among the cluster RCTs, six were considered at unclear risk of recruitment bias as there was no explicit

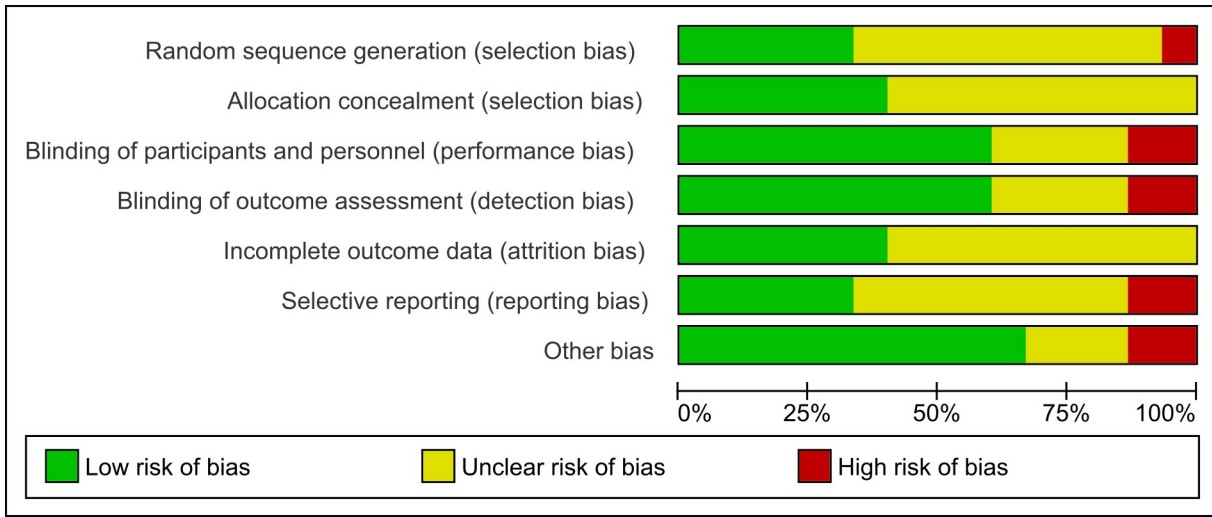

**Fig 2. Risk of bias graph: Review authors' judgements about each risk of bias item presented as percentages across all included studies.**

information on whether the individuals were recruited to the trial after the clusters have been randomized [28–31,34,36]. One trial was at high risk of baseline imbalance, as the baseline mortality in children was slightly different between the intervention and control arm [28]. Regarding incorrect analysis in cluster RCTs, i.e. not taking the clustering into account, most studies have considered the effect of clustering except for a few studies where we could not find any information about cluster adjustment [28, 36]. See "S3 Appendix" for details.

## Effects of intervention

**All-cause mortality at longest follow-up.** We incorporated 15 trials with a total of around 1,200,679 children in the meta-analysis for the outcome of all-cause mortality at the longest follow-up. The results showed a considerable qualitative and quantitative heterogeneity, with an overall small protective effect (RR 0.89, 95% CI 0.84 to 0.94; $Chi^2$ = 48.87, degrees of freedom (df) = 14; P < 0.00001; $I^2$ = 71% Fig 3) of VAS resulting in 11% reduction in mortality, with half the weight attributed to the large DEVTA study.

This effect estimate is almost the same as the estimate reported in a previous review (RR 0.88, 95% CI 0.83 to 0.93) [6]. Due to the heterogeneity, we also conducted the analysis using a random-effects model, producing a differing effect estimate (RR 0.78, 95% CI 0.68 to 0.91, heterogeneity: $Tau^2$ = 0.04; $Chi^2$ = 48.87, df = 14; P < 0.00001; $I^2$ = 71% See "S1 Fig"). Given the heterogeneity, we first investigated if this could be explained by the risk of bias, and then explored the pre-specified effect modifiers.

## Sensitivity analyses

**Bias.** Of the included studies, one study had a high risk of bias for sequence generation [30] but only accounted for 4.8% of the weight and did not influence the effect. More than half (60%) of the included studies had an unclear risk of bias for allocation concealment. Stratified analysis of the studies by inadequate allocation concealment (RR 0.84, 95% CI 0.76 to 0.93; $I^2$:70%) and adequate allocation concealment (RR 0.91, 95% CI 0.85 to 0.97; $I^2$: 75%; 6 trials, 1,046,829 participants See Fig 4) suggested that at least some of the effect estimate was the result of bias. We, therefore, used the effect estimate of the higher-quality studies for the primary analysis.

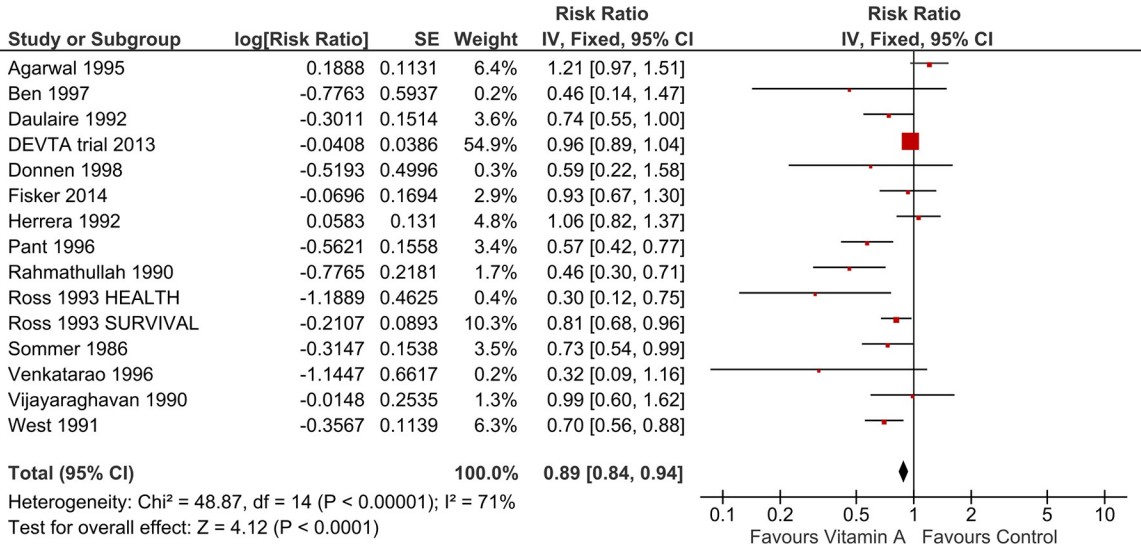

**Fig 3. Forest plot of comparison: 1 Vitamin A versus Control, outcome: All-cause mortality at longest follow-up.**

| Study or Subgroup | log[Risk Ratio] | SE | Weight | Risk Ratio IV, Fixed, 95% CI |
|---|---|---|---|---|
| **1.2.1 Adequate allocation concealment** | | | | |
| Ben 1997 | -0.7763 | 0.5937 | 0.2% | 0.46 [0.14, 1.47] |
| DEVTA trial 2013 | -0.0408 | 0.0386 | 54.9% | 0.96 [0.89, 1.04] |
| Fisker 2014 | -0.0696 | 0.1694 | 2.9% | 0.93 [0.67, 1.30] |
| Rahmathullah 1990 | -0.7765 | 0.2181 | 1.7% | 0.46 [0.30, 0.71] |
| Ross 1993 HEALTH | -1.1889 | 0.4625 | 0.4% | 0.30 [0.12, 0.75] |
| Ross 1993 SURVIVAL | -0.2107 | 0.0893 | 10.3% | 0.81 [0.68, 0.96] |
| Subtotal (95% CI) | | | 70.4% | 0.91 [0.85, 0.97] |

Heterogeneity: Chi² = 20.35, df = 5 (P = 0.001); I² = 75%
Test for overall effect: Z = 2.74 (P = 0.006)

| | | | | |
|---|---|---|---|---|
| **1.2.2 Inadequate allocation concealment** | | | | |
| Agarwal 1995 | 0.1888 | 0.1131 | 6.4% | 1.21 [0.97, 1.51] |
| Daulaire 1992 | -0.3011 | 0.1514 | 3.6% | 0.74 [0.55, 1.00] |
| Donnen 1998 | -0.5193 | 0.4996 | 0.3% | 0.59 [0.22, 1.58] |
| Herrera 1992 | 0.0583 | 0.131 | 4.8% | 1.06 [0.82, 1.37] |
| Pant 1996 | -0.5621 | 0.1558 | 3.4% | 0.57 [0.42, 0.77] |
| Sommer 1986 | -0.3147 | 0.1554 | 3.4% | 0.73 [0.54, 0.99] |
| Venkatarao 1996 | -1.1447 | 0.6617 | 0.2% | 0.32 [0.09, 1.16] |
| Vijayaraghavan 1990 | -0.0148 | 0.2535 | 1.3% | 0.99 [0.60, 1.62] |
| West 1991 | -0.3567 | 0.1139 | 6.3% | 0.70 [0.56, 0.88] |
| Subtotal (95% CI) | | | 29.6% | 0.84 [0.76, 0.93] |

Heterogeneity: Chi² = 26.76, df = 8 (P = 0.0008); I² = 70%
Test for overall effect: Z = 3.34 (P = 0.0008)

| | | | | |
|---|---|---|---|---|
| **Total (95% CI)** | | | **100.0%** | **0.89 [0.84, 0.94]** |

Heterogeneity: Chi² = 48.84, df = 14 (P < 0.00001); I² = 71%
Test for overall effect: Z = 4.12 (P < 0.0001)
Test for subgroup differences: Chi² = 1.73, df = 1 (P = 0.19), I² = 42.2%

**Fig 4. Forest plot of comparison: 1 Vitamin A versus control, outcome: 1.2 All-cause mortality (sensitivity analysis by allocation concealment).**

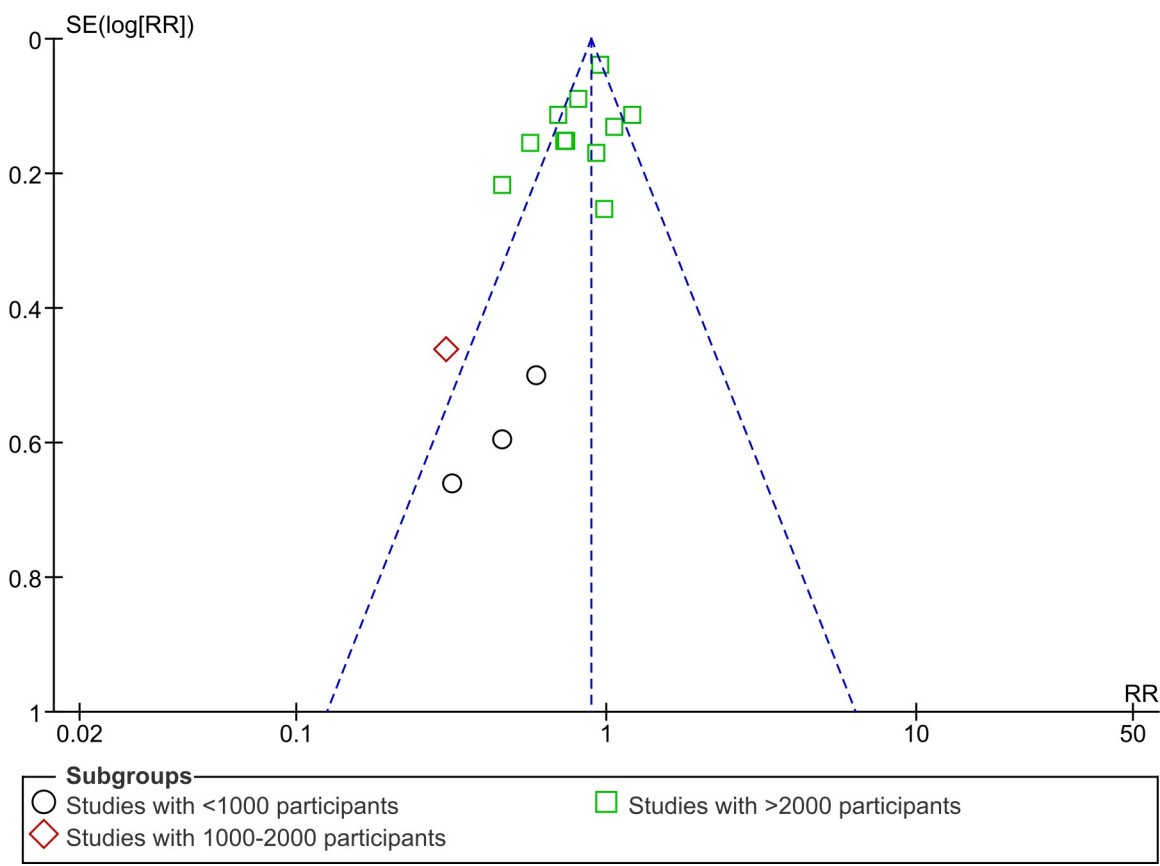

**Fig 5. Funnel plot of comparison: 1 Vitamin A versus control, outcome: All-cause mortality at longest follow-up (sub-group analysis by number of participants in each study).**

**Funnel plot.** We generated a funnel plot (Fig 5) for the outcome of mortality based on the number of participants in each study. We categorized the included studies as small studies having less than 1000 participants and between 1000–2000 participants and large studies having greater than 2000 participants. We can see an asymmetrical funnel plot in Fig 5 with no small studies favouring control. While some studies are showing a protective effect in the bottom-left corner, small studies in the bottom-right corner remained unaccounted for. Also, the large protective effects of Vitamin A are mainly demonstrated by smaller studies. It points to the likely presence of small-study effects or publication bias which might have played a role in the overestimation of the mortality reduction effect estimate.

**Sub-group analyses.** Given the substantial heterogeneity (P-value <0.10 and I2 > 50%) we conducted different sub-group analysis. For details of the sub-group analysis conducted, See "S2–S5 Figs".

**Sub-group analysis by decade.** 13 studies conducted before 2000 reported about 20% reduction (RR 0.80, 95% CI 0.74 to 0.87; $Chi^2$ = 39.26, degrees of freedom (df) = 12; P < 0.00001; $I^2$ = 69%; Fig 6) in mortality associated with VAS [23,25–34,36,37]. In the two studies conducted after 2000, the effect estimate (RR 0.96, 95% CI 0.89 to 1.03; $Chi^2$ = 0.03, degrees of freedom (df) = 1; P = 0.87; $I^2$ = 0%; 1,007,587 participants) suggested 4% reduction in overall child mortality [24,35]. This 4% reduction, however, included the possibility of both a reduction and an increase in the risk of mortality with Vitamin A. This suggests that the effects from trials conducted over 20 years ago showed some substantive effects and other smaller effects. The 20% reduction is in the presence of substantive heterogeneity, which might

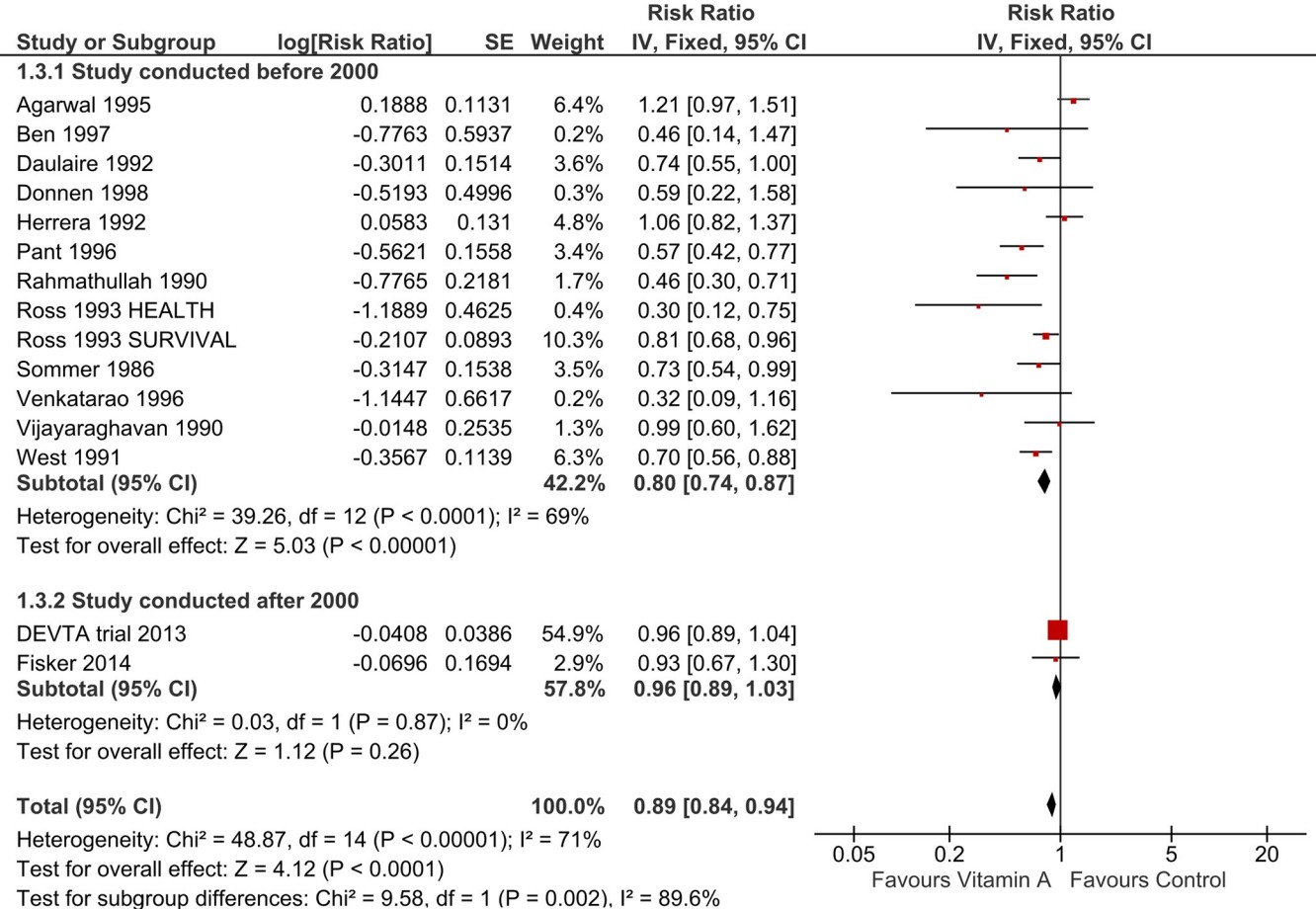

**Fig 6. Forest plot of comparison: Vitamin A versus control, Outcome: All-cause mortality at longest follow-up (sub-group analysis by decade).**

be expected given deficiency is likely to vary. There is no contemporary (last ten years) evidence of an effect, but only two studies are being conducted during this period of ten years.

**Sub-group analysis by background/baseline Under-Five Mortality Rate (U5MR).** Sub-group analysis stratified by U5MR in the primary studies showed smaller effects (RR 0.96, 95% CI 0.90 to 1.03) when the background U5MR ranged between 90-120/1000 live births compared to when U5MR was greater than 120/1000 live births (RR 0.75, 95% CI 0.68 to 0.83) (See Fig 7). Mortality estimates were not possible to estimate across the sub-groups (U5MR: 30-60/1000 live births and U5MR: 60-90/1000 live births) as none of the included studies had baseline U5MR between 30-90/1000 live births. The background U5MR in all the included trials in the meta-analysis was greater than 90 per 1000 live births and none of the trials had U5MR close to the current U5MR (28/1000 live births) in Nepal. The present mortality in Nepal is almost three times less than the levels reported in those trials. So, there are no trials that could represent the current under-five mortality context of Nepal.

**Potential Impact of Vitamin A supplementation in Nepal.** We estimated the potential effects of supplementation in Nepal by applying the effect estimates from the review to the current U5MR in Nepal. For this, we used the best estimate of effect obtained from high quality adequately concealed studies to current values of U5MR in Nepal. We also appraised the quality of the evidence using GRADE methods. The certainty of the evidence is of low quality suggesting that VAS may reduce mortality in children. As indicated in Table 3, when the relative

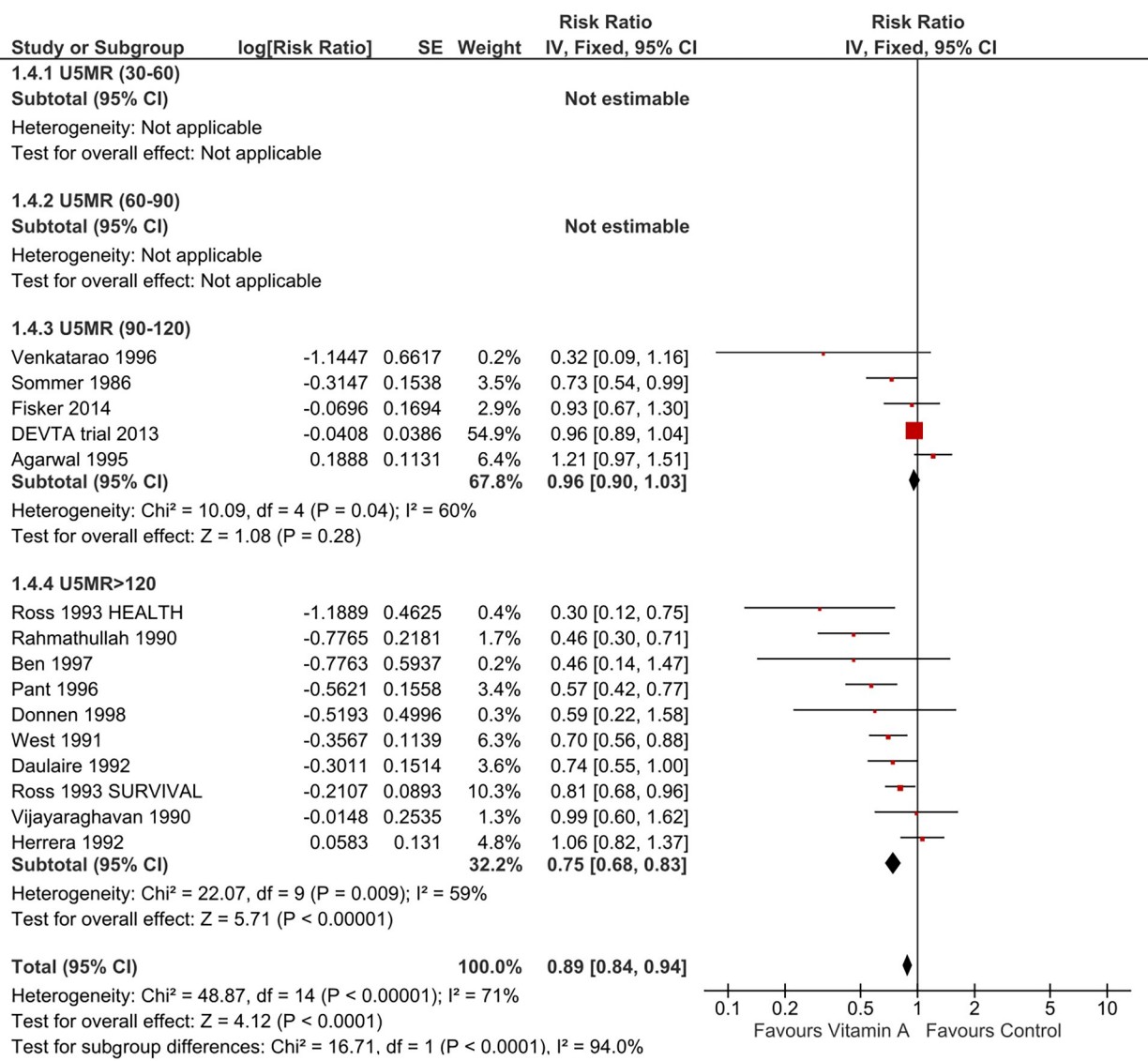

**Fig 7. Forest plot of comparison: Vitamin A versus control, outcome: All-cause mortality at longest follow-up (sub-group analysis by background U5MR).**

risk reduction estimate was applied to the national U5MR in Nepal, it resulted in a reduction in mortality from 28 per 1000 live births to 25 per 1000 live births (Low-quality evidence). The absolute benefit of supplementation is small (0.3%) as Vitamin A may only reduce 3 deaths per 1000 children supplemented with Vitamin A (See Table 3). Even in the province where U5MR is as high as 40 per 1000 live births, there would only be a small reduction in mortality to 36 per 1000 live births (low-quality evidence) (See "S1 Table" for details).

## Discussion

### Summary of main results

In summary, VAS resulted in a small protective effect against child mortality with most recent trials conducted after 2000 showing small VAS benefits. Evidence from stratified analysis by allocation concealment and U5MR indicated little to no difference in mortality reduction with VAS.

**Table 3. Summary of Findings using the GRADE methods for estimating the effects of Vitamin A supplementation in Nepal.** Applying the best estimate of effect to national statistics of Nepal for Under-Five Mortality Rate.

| Outcomes | Anticipated absolute effects* (95% CI) | | Relative effect (95% CI) | Absolute (95% CI) | № of participants (studies) | Certainty of the evidence (GRADE) | Comments |
|---|---|---|---|---|---|---|---|
| | Risk with control (current estimates) | Risk with Vitamin A supplementation | | | | | |
| **All-cause child mortality** | **28 per 1,000** (U5MR in Nepal) | **25 per 1,000** (24 to 27) | **RR 0.91** (0.85 to 0.97) | **3 fewer per 1,000** (from 4 fewer to 1 fewer) | 1,046,829 (6 RCTs) | ⊕⊕◯◯ Low[a,b] | Vitamin A supplementation may result in a small reduction in child mortality |

*Explanations*.

a. Downgraded 1 level due to serious imprecision (Effect estimate includes both negligible effect (3% reduction) and considerable benefit (15% reduction) with vitamin A supplementation).

b. Downgraded 1 level due to serious inconsistency ($I^2$ was 75%, and the results of Rahmathullah 1990; Ross 1993 HEALTH and Ross 1993 SURVIVAL demonstrated evidence of benefit contrary to the results of other studies).

*The risk in the intervention group** (and its 95% confidence interval) is based on the assumed risk in the comparison group and the **relative effect** of the intervention (and its 95% CI).

**CI:** Confidence Interval; **RR:** Risk ratio **U5MR:** Under-Five Mortality Rate.

**GRADE Working Group grades of evidence**.

**High certainty**: We are very confident that the true effect lies close to that of the estimate of the effect.

**Moderate certainty**: We are moderately confident in the effect estimate: The true effect is likely to be close to the estimate of the effect, but there is a possibility that it is substantially different.

**Low certainty**: Our confidence in the effect estimate is limited: The true effect may be substantially different from the estimate of the effect.

**Very low certainty**: We have very little confidence in the effect estimate: The true effect is likely to be substantially different from the estimate of effect.

The decrease in the effect of VAS can be linked to the background parameters in the trials and secular changes that occurred over the period of time. If we look into the background parameters for the trials, trials conducted before 2000 featured high U5MR, greater levels of wasting and xerophthalmia and low coverage of measles immunization, while the trials after 2000 reported slightly improved status for these indicators. This might explain the beneficial effect of VAS that was seen in the trials conducted before 2000 when the prevalence of xerophthalmia was much higher. Results from this review corroborate the findings from the previous review [6]. However, the absolute benefits from VAS are small in reducing mortality and the latest effect estimate from the DEVTA trial, which occupies the largest weight in the analysis, did not show any VAS benefits in recent times [35]. Thus the conclusion of the Cochrane review that "Vitamin A supplementation is associated with a clinically meaningful reduction in morbidity and mortality in children" is probably not correct: we recommend national policy makers use the estimates from the review applied to their national data to determine themselves the potential public health benefits of continuing routine supplementation.

## Certainty of the evidence (GRADE analysis) and its impact in Nepal

The overall certainty of the best estimate of effect obtained from the well-concealed studies is of low quality. We downgraded the certainty rating of the evidence from high to low due to concerns related to the consistency of the estimate (qualitative and quantitative heterogeneity) and precision (95% CI includes the possibility of both negligible reduction and appreciable important reduction in mortality with VAS).

The potential impact of VAS is found to be low as VAS may result in a little reduction in mortality in absolute terms with only three deaths prevented for every 1000 children supplemented with VAS.

Given the lower absolute benefit of VAS, the relevance of routine VAS for reducing child mortality in the present context in Nepal raises doubts. As evidenced from the analysis, no trials included in the review reflected the current status of mortality rates and other indicators of population health in Nepal. Also, the presence of heterogeneity with small studies reporting larger effects and recent large studies [35] occupying major weight in the analysis reporting no effect limited our confidence in the certainty regarding the potential impact that VAS may have in Nepal in reducing child mortality.

## Conclusions

The overall benefit of VAS in reducing child mortality in Nepal is small in absolute terms. There is probably sufficient evidence to cease current routine supplementation programs.

## Supporting information

**S1 Checklist. PRISMA 2020 checklist.**
(DOCX)

**S1 Fig. Forest plot of comparison: 1 Vitamin A versus Control, outcome: 1.2 All-cause mortality at longest follow-up (random-effect model).**
(TIFF)

**S2 Fig. Sub-group analysis by wasting.**
(TIFF)

**S3 Fig. Sub-group analysis by xerophthalmia.**
(TIFF)

**S4 Fig. Sub-group analysis by measles immunization coverage.**
(TIFF)

**S5 Fig. Sub-group analysis by vitamin A coverage.**
(TIFF)

**S1 Table. Using the GRADE methods for estimating effects of Vitamin A supplementation in Nepal at the sub-national level.**
(DOCX)

**S1 Appendix. Methods.**
(DOCX)

**S2 Appendix. Characteristics of excluded studies.**
(DOCX)

**S3 Appendix Characteristics of included studies.**
(DOCX)

## Acknowledgments

We would like to acknowledge the national and international nutrition experts and researchers who helped in the conceptualization of this study. The authors would like to thank Professor Peter Heywood, Honorary Research Associate, the University of Sydney, and Dr. Marianne Visser, University of Stellenbosch, South Africa for their invaluable input on the review protocol.

## Author Contributions

**Conceptualization:** Samjhana Shrestha, Saki Thapa, Paul Garner, Maxine Caws, Suman Chandra Gurung, Khem Narayan Pokhrel.

**Data curation:** Samjhana Shrestha, Richard Kirubakaran, Khem Narayan Pokhrel.

**Formal analysis:** Samjhana Shrestha, Paul Garner, Khem Narayan Pokhrel.

**Funding acquisition:** Saki Thapa, Paul Garner, Maxine Caws, Suman Chandra Gurung.

**Investigation:** Samjhana Shrestha, Saki Thapa, Paul Garner, Maxine Caws, Suman Chandra Gurung, Khem Narayan Pokhrel.

**Methodology:** Samjhana Shrestha, Saki Thapa, Paul Garner, Maxine Caws, Suman Chandra Gurung, Richard Kirubakaran, Khem Narayan Pokhrel.

**Project administration:** Samjhana Shrestha, Saki Thapa, Paul Garner, Maxine Caws, Suman Chandra Gurung, Tilly Fox, Khem Narayan Pokhrel.

**Resources:** Samjhana Shrestha, Saki Thapa, Paul Garner, Maxine Caws, Khem Narayan Pokhrel.

**Software:** Samjhana Shrestha, Paul Garner, Tilly Fox, Richard Kirubakaran, Khem Narayan Pokhrel.

**Supervision:** Saki Thapa, Paul Garner, Maxine Caws, Suman Chandra Gurung, Khem Narayan Pokhrel.

**Validation:** Samjhana Shrestha, Saki Thapa, Paul Garner, Maxine Caws, Tilly Fox, Khem Narayan Pokhrel.

**Visualization:** Samjhana Shrestha, Paul Garner, Tilly Fox, Richard Kirubakaran, Khem Narayan Pokhrel.

**Writing – original draft:** Samjhana Shrestha.

**Writing – review & editing:** Samjhana Shrestha, Saki Thapa, Paul Garner, Maxine Caws, Suman Chandra Gurung, Tilly Fox, Richard Kirubakaran, Khem Narayan Pokhrel.

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
