## [Decision Letter · Decision Letter 0]

29 Mar 2022

PONE-D-21-38653Is routine Vitamin A still justified for children in Nepal?  Evidence analysis using contemporary estimatesPLOS ONE

Dear Dr. Pokhrel,

Thank you for submitting your manuscript to PLOS ONE. We feel that your manuscript has merit but does not fully meet PLOS ONE’s publication criteria as it currently stands. Therefore, we invite you to submit a revised version of the manuscript that addresses the points raised during the review process, especially the comments by the reviewer regarding the difference between the recent Cochrane review and your results.

We look forward to receiving your revised manuscript.

Kind regards,

Frank Wieringa, M.D., Ph.D.

Academic Editor

PLOS ONE

Journal Requirements:

Reviewers' comments:

Reviewer's Responses to Questions

**Comments to the Author**

1. Is the manuscript technically sound, and do the data support the conclusions?

Reviewer #1: Yes

2. Has the statistical analysis been performed appropriately and rigorously? 

Reviewer #1: Yes

3. Have the authors made all data underlying the findings in their manuscript fully available?

Reviewer #1: Yes

4. Is the manuscript presented in an intelligible fashion and written in standard English?

Reviewer #1: Yes

5. Review Comments to the Author

Reviewer #1: This study re-examines the effect of vitamin A supplementation on under 5 mortality in the context of current mortality estimates in Nepal.

This is a meta analysis of a subset of studies earlier examined in the Cochrane review by Imdad et al. It is very relevant to revisit the need for supplementation in Nepal as the vit A supplementation is an ongoing program which has not been evaluated. Therefore the paper provides important insights in this area. The paper introduces the need for the study clearly.

1. The title states that “evidence analysis using contemporary estimates”. The authors attempted to update the earlier meta analysis with more recent studies, However, there was only one study which was not included in the earlier meta analysis, Eventually this one additional study was also dropped from the meta-analysis because it did not fulfil the criteria of inclusion. Therefore although the authors claim review of recent studies, eventually there are no more recent studies included in the Cochrane review and the title may not be justified.

2. This study chose only a subset of the studies in cluded in the Cochrane review. The criteria for selection of studies for re-analysis is not clear and it seems to have not considered a list of studies without clear justification. Eg: Donnen 1998 estimate of all cause mortality is presented in the Cochrane review although this study is dropped from the current analysis. The paper by Donnen et al did report 5.4% mortality in the Vit A group compared to 9% in control group resulting in a risk ratio of 0.6 which is reported in the Cochrane review. This was not a primary outcome of the study, but this is not sufficient justification for dropping the study. Hence the exclusion of studies should be re-considered.

3. The authors state that no study has a baseline U5mortality estimate close to that of Nepal which is 28 in 1000. But the next line states that “Thus, the mortality estimate most similar to current values in Nepal gives an estimate of 4% (RR 0.96, 95% CI 0.90 to 1.03) reduction in child mortality with VAS.” Which study is represented here?

4. It might be useful to do a separate analysis of the Nepali studies considered in the Cochrane review as has been done in the recently published Indian Pediatrics paper (https://www.indianpediatrics.net/epub092021/RP-00372.pdf)

6. PLOS authors have the option to publish the peer review history of their article (what does this mean?). If published, this will include your full peer review and any attached files.

Reviewer #1: No

---

## [Author Response · Author response to Decision Letter 0]

11 Apr 2022

Dear Editor and Reviewers,

Thank you for taking time to review our manuscript. We have revised the manuscript and submitted revised version for your review.

---

## [Editor Report · Decision Letter 1]

3 May 2022

Is routine Vitamin A supplementation still justified for children in Nepal? Trial synthesis findings applied to Nepal national mortality estimates

PONE-D-21-38653R1

Dear Dr. Pokhrel,

We’re pleased to inform you that your manuscript has been judged scientifically suitable for publication and will be formally accepted for publication once it meets all outstanding technical requirements.

Please delete, in line 75 'a' before 'decade'.

Kind regards,

Frank Wieringa, M.D., Ph.D.

Academic Editor

PLOS ONE
---

## [Editor Report · Acceptance letter]

10 May 2022

PONE-D-21-38653R1 

Is routine Vitamin A supplementation still justified for children in Nepal? Trial synthesis findings applied to Nepal national mortality estimates 

Dear Dr. Pokhrel:

I'm pleased to inform you that your manuscript has been deemed suitable for publication in PLOS ONE. Congratulations! Your manuscript is now with our production department. 

Kind regards, 

on behalf of

Dr. Frank Wieringa 

Academic Editor

PLOS ONE